EMBO
Molecular Medicine

# Organelle resilience as a comparative blueprint for longevity

Domagoj Cikes ORCID ✉

## Abstract

The past decade has defined molecular hallmarks of aging, yet interventions that extend lifespan in short-lived organisms show limited and context-dependent translation to humans. Comparative studies of exceptional longevity remain largely genome-centric, although genomic instability alone cannot comprehensively explain aging-related pathologies. Many age-associated failures emerge at the level of cellular organelles whose stability underpins tissue function. The pathways that sustain these structures operate through proteomic, metabolic, and lipid networks that are insufficiently captured by genomic or transcriptomic analyses. Notably, longer organismal lifespan increases the requirement for sustained organelle functionality and fidelity. This Perspective proposes that the next conceptual advance in geroscience will come from comparative organelle biology. Examining mammals with divergent lifespans, including species evolutionarily closer to humans, can reveal how long-lived lineages evolved organelle-level architecture and resilience mechanisms that support cellular function over decades. I introduce the Comparative Metabolic Longevity Cell Atlas (CMLCA), a cross-mammalian platform integrating standardized cellular systems, organelle-resolved multi-omics, and computational analysis to identify conserved features of resilience and inform next-generation strategies to improve human healthspan.

## Background

Over the past decades, aging research has identified conserved molecular pathways that influence lifespan across species, leading to the formulation of widely recognized hallmarks of aging and establishing a mechanistic foundation for geroscience (Schmauck-Medina et al, 2022; López-Otín et al, 2023). These include genomic instability, epigenetic alterations, loss of proteostasis, deregulated nutrient sensing, mitochondrial dysfunction, cellular senescence, stem cell exhaustion, and altered intercellular communication, which together describe key molecular and cellular drivers of aging. Many of these pathways were first discovered in simple model organisms and later validated in mammals, underscoring strong evolutionary conservation. Yet translating these insights into consistent benefits for human aging remains complex, with outcomes varying across tissues, physiological contexts, and intervention strategies (Bjelakovic et al, 2014; Moel et al, 2025; Dollerup et al, 2018; Orr et al, 2024; Shoji et al, 2025; Brakedal et al, 2022; Yi et al, 2023; Yoshino et al, 2021; Denk et al, 2025; Bruyn et al, 2008; Mannick et al, 2014; Chung et al, 2019; Krebs et al, 2007; Palma et al, 2022; Singh et al, 2022; Liu et al, 2022; Steinbrücker et al, 2023; Veenhuis et al, 2021; Yulug et al, 2023; Fang et al, 2019; Lee et al, 2024).

Comparative studies of long-lived mammals have expanded this framework but remain largely centered on genomic variation, DNA repair, and cancer resistance (Firsanov et al, 2025; Foley et al, 2018; Chen et al, 2025; Sulak et al, 2016). While essential for defining regulatory potential, these layers correlate only partially with the proteostatic, lipid, and metabolic networks that are critical to sustain cellular and tissue function to stave off multiple age-related diseases. In addition, many exceptionally long-lived species occupy ecological niches distinct from those of humans, complicating direct inference of mechanisms relevant to human aging biology.

Many age-associated dysfunctions emerge at the level of cellular organelles, whose long-term stability underpins tissue function. The processes sustaining these compartments operate through proteomic, metabolic, and lipid networks that are incompletely captured by genome-centric analyses. Their functional demands also differ markedly across species——with organelles maintaining homeostasis for years in short-lived organisms, such as mice, and for decades in long-lived, such as primates and humans, reflecting evolutionary pressures on long-term organellar functionality and resilience in long-lived species. Accordingly, the kinetics of organelle turnover, metabolic tempo, and damage handling vary across lifespan scales and may influence the effectiveness of interventions developed primarily in short-lived species.

In this Perspective, I propose that the next conceptual advance in geroscience will come from comparative organelle biology. Longevity in long-lived species is likely shaped by organelle-level resilience mechanisms adapted to extended lifespan. Identifying these mechanisms, particularly in mammals closer to humans, such as the primates, may reveal experimentally and pharmacologically tractable pathways more compatible with human cellular physiology.

I introduce the developing Comparative Metabolic Longevity Cell Atlas (CMLCA)——an emerging cross-mammalian platform that enables systematic comparative analysis of standardized cellular systems with organelle-resolved multi-omics. By uncovering conserved molecular architectures shaped by evolutionary pressures for long-term stability, this framework aims to inform more predictive strategies for improving human healthspan.

Division of Endocrinology and Metabolism, Department of Medicine III, Medical University of Vienna, Vienna, Austria. ✉E-mail: Domagoj.cikes@meduniwien.ac.at
https://doi.org/10.1038/s44321-026-00428-2 | Published online: 16 April 2026

   

## Promise and limitations of current translational geroscience

### Landscape of anti-aging therapies: promise and limitations

Efforts to delay aging in humans have been shaped by non-pharmacological and pharmacological interventions. Among non-pharmacological interventions, reduction of nutrient intake, including caloric restriction and fasting, remains one of the most consistent modulators of lifespan across species, highlighting the central role of metabolic state in aging biology.

Concerning current pharmacological interventions, these often show robust benefits in short-lived organisms but currently often yield modest, variable, or context-dependent outcomes in humans (Fig. 1). Metformin is one of the most discussed geroprotective drugs which in preclinical models showed robust effects on lifespan and/or on healthspan by mimicking fasting (Martin-Montalvo et al, 2013; Yang et al, 2024; Chen et al, 2017). Epidemiology and small mechanistic trials are encouraging, and the large TAME trial is planned to test delayed multimorbidity in non-diabetic older adults, but it is still in preparation,

and no definitive prospective evidence exists (Du et al, 2022). Nicotinamide adenine dinucleotide (NAD⁺) has emerged as a major target in translational geroscience (Zhang et al, 2025). As a central cofactor in mitochondrial metabolism, redox balance, and DNA repair, NAD⁺ supports organelle function and cellular homeostasis, and its decline with age has been proposed to contribute to dysfunction in metabolically demanding tissues such as muscle, brain, and kidney (Yusri et al, 2025). In preclinical systems, augmentation of NAD⁺ through precursors such as nicotinamide riboside (NR) and nicotinamide mononucleotide

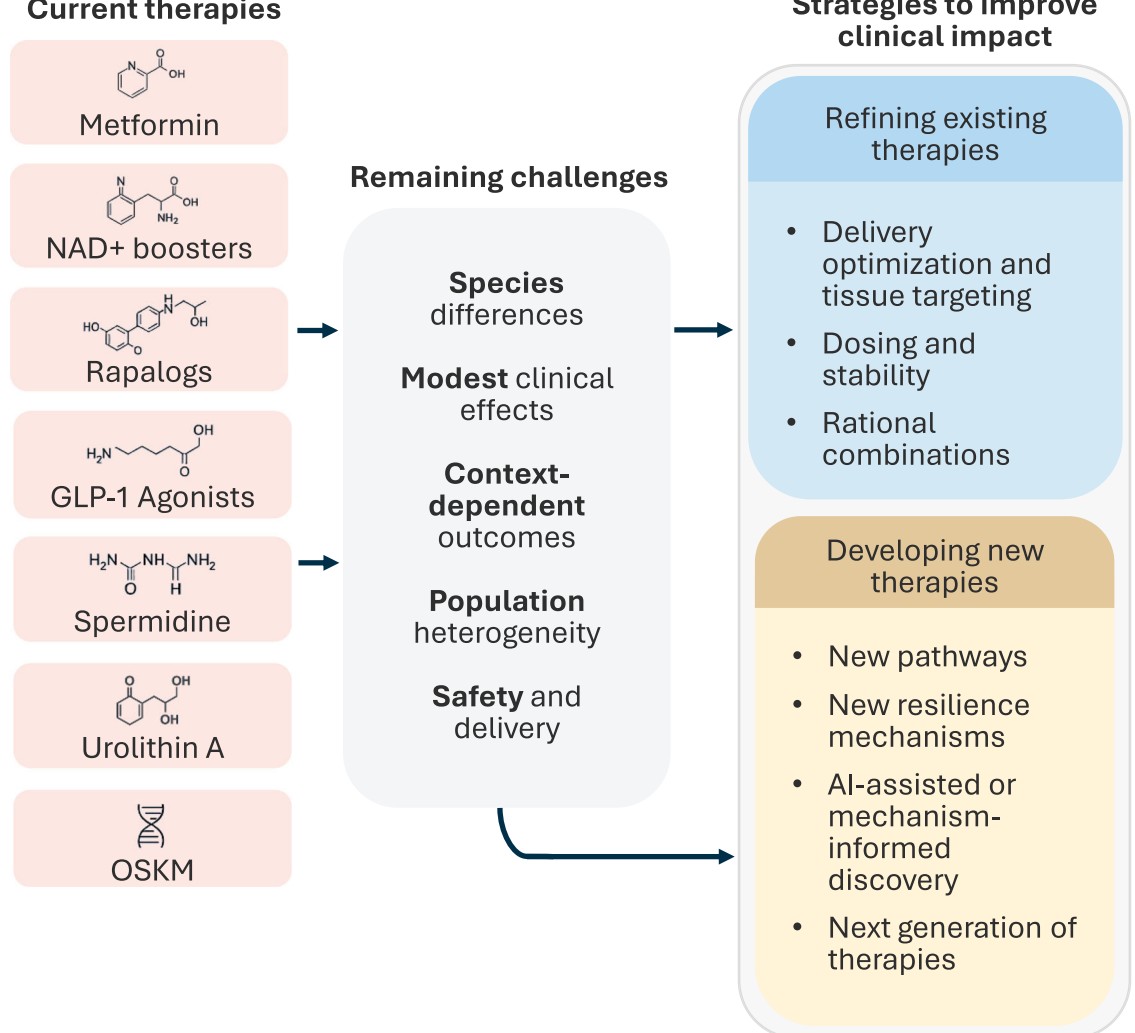

**Figure 1. Current anti-aging interventions: translational challenges and strategies for clinical improvement.**

Current therapeutic classes, remaining translational challenges, and strategies to improve clinical impact. Currently tested interventions—including metabolic modulators, nutrient-sensing inhibitors, and partial reprogramming approaches—show robust effects in model organisms but modest or inconsistent outcomes in humans. Key barriers include species differences, context-dependent responses, population heterogeneity, and delivery or safety constraints. Improving clinical efficacy will require both optimization of existing modalities (e.g., dosing, targeting, stability) and discovery of next-generation interventions informed by resilience pathways and AI-enabled prediction.

(NMN) produces improvements of multiple age-related pathologies in models of aging (Mills et al, 2016) and neurodegeneration (Wang et al, 2016), as well as in genetically predisposed premature-aging contexts (Fang et al, 2016), without any obvious toxicity or deleterious effects, supporting a mechanistic rationale for therapeutic NAD⁺ restoration. In humans, NAD⁺ precursors reproducibly and safely increase circulating NAD⁺ and related metabolites (Freeberg et al, 2023). Early-phase studies indicate biological and metabolic activity across metabolic, cardiovascular, and neurodegenerative contexts (Zhang et al, 2025). Measurable benefits have been reported in defined settings, including improved insulin sensitivity in prediabetic individuals (Yoshino et al, 2021), enhanced walking performance and muscle function in older adults (Igarashi et al, 2022), and symptomatic improvements (Orr et al, 2024, 2020) in neurodegenerative disorders in subsets of patients (Brakedal et al, 2022). Beneficial effects have also been observed in genetically predisposed premature-aging disorders (Veenhuis et al, 2021; Steinbrücker et al, 2023) and cardiometabolic risk profiles (Prineas and Friedewald, 1986). Based on current short-term trials, functional outcomes appear to depend on baseline metabolic state, tissue vulnerability, and treatment duration (Freeberg et al, 2023). Collectively, these findings suggest that NAD⁺ augmentation may restore aspects of tissue and organelle function under specific conditions, while its broader impact on long-term resilience in normal aging remains to be determined.

Another widely studied class of interventions targets the evolutionarily conserved mTOR pathway. Rapamycin and related mTOR inhibitors represent among the most reproducible pharmacological modulators of aging identified to date, extending lifespan and healthspan across multiple model organisms and in numerous mouse studies (Kennedy and Lamming, 2016; Harrison et al, 2009; Liao et al, 2016), where it improves a broad spectrum of age-associated pathologies across cardiovascular, immune, musculoskeletal, metabolic, nervous and reproductive systems (Siddiqui et al, 2015; Garcia et al, 2019; Martínez-Cisuelo et al, 2016; Hurez et al, 2015), positioning mTOR modulation as a leading pharmacological candidate in geroscience. In humans, early trials indicate biological activity at lower or intermittent dosing, including improvements in immune function, glucose uptake, vaccine responsiveness, and skin aging (Bruyn et al, 2008; Mannick et al, 2014; Chung et al, 2019; Krebs et al, 2007), supporting relevance beyond preclinical systems. Studies in older adults further suggest potential benefits for muscle mass, self-reported general health measures, and well-being with minimal side effects under carefully controlled dosing regimens (Moel et al, 2025). At the same time, long-term application in aging populations remains under active investigation. Existing clinical studies are relatively short and involve limited cohorts, and higher or continuous dosing can produce adverse metabolic and immunological effects, underscoring the importance of dose, timing, and treatment strategy (Lee et al, 2024). This example further highlights the need to better understand how systemic conserved pathway targeting interfaces with species-specific cellular metabolism. A similar pattern is evident across other intervention classes. Senolytics reverse aging features in mice across many tissues (Fuhrmann-Stroissnigg et al, 2017; Chang et al, 2016; Baar et al, 2017), yet human studies have produced variable results, with concerns about specificity and off-target toxicity (de Magalhães, 2025). GLP-1 receptor agonists (e.g., semaglutide) strongly improve metabolic health and cardiovascular outcomes (Marso et al, 2016; Zelniker et al, 2019), with early hints of neuroprotective effects (Athauda et al, 2017), but their relevance to aging in metabolically healthy individuals remains unknown. The polyamine spermidine promotes autophagy and mitochondrial turnover (Eisenberg et al, 2009); higher dietary intake correlates with reduced mortality, and early trials suggest cognitive benefits, but causal evidence is still limited (Wirth et al, 2018; Schwarz et al, 2020). Similarly, urolithin A shows robust effects in preclinical studies (Ryu et al, 2016; D'Amico et al, 2021), and early clinical trials report no adverse effects together with modest improvements in mitochondrial activity, autophagy, inflammation, and muscle performance; however, no effects on overall physical function have been observed, underscoring the need for larger, longer, and more multi-system human studies, as evidence for urolithin A efficacy in human aging remains limited (Kuerec et al, 2024). Partial cellular reprogramming produces striking rejuvenation in progeria models (Ocampo et al, 2016) and aged human cells (Sarkar et al, 2020), yet safety, delivery, and efficacy in normal aging remain major open questions, with the strongest effects so far in accelerated-aging contexts (Browder et al, 2022; Sarkar et al, 2020). These advances have begun to translate toward clinical testing: early regulatory clearance has enabled first-in-human studies of partial epigenetic reprogramming approaches, including therapies based on transient delivery of OSK factors (OCT4, SOX2, KLF4) for age-related conditions such as optic neuropathies. Despite this progress, major questions remain regarding safety, delivery, durability, and efficacy in normal aging. To date, the most pronounced effects have been observed in accelerated-aging or disease contexts, indicating that cellular state, tissue vulnerability, and underlying damage burden strongly influence responsiveness to reprogramming interventions. Overall, translating promising interventions into stronger and broader human benefits remains challenging, likely reflecting evolutionary differences in tissue physiology, metabolic tempo, and the long-term maintenance demands placed on cellular systems. Together, these observations underscore the need to understand how longevity interventions operate across different biological scales and lifespans, particularly when translating findings from experimental models to humans.

## Constraints of current aging models

A major challenge for clinical translation arises from differences in biological scale and cellular context across experimental systems. Foundational discoveries in aging have emerged from model organisms spanning large evolutionary distances from humans, including yeast, nematodes, insects, and short-lived vertebrates. These systems have been essential for identifying conserved pathways regulating longevity, yet they operate under distinct metabolic regimes, tissue architectures, and lifespan tempos that shape how aging processes unfold (Fig. 2A). *Saccharomyces cerevisiae* (>1 billion years of divergence) relies predominantly on fermentative rather than oxidative metabolism even under aerobic conditions (Maslanka et al, 2020), limiting its relevance to mammalian mitochondrial biology. *Caenorhabditis elegans* (~600 Myr divergence) possesses a largely post-mitotic soma and a simplified insulin/IGF-like system with a single IIS receptor (DAF-2)

## A

**Classical Translational Paradigm**

**Classical models**

**Advantages:**
- Short-lifespan
- Fast evaluation of therapeutic efficacy
- Genetic tractability
- Conserved pathways

**Disadvantages:**
- Structures evolved for short duration
- Limited physiological complexity
- Limited pathway complexity

1 billion to 90 million years of evolutionary distance

**Translational bottleneck**

Divergent physiology

Different mechanisms of degeneration

Limited predictive power

Need for expanded model systems

## B

**Comparative biology paradigm**

**Exceptional long-lived species**

**Advantages:**
- Mammalian physiology
- High tissue/pathway complexity
- Conserved molecular pathways

**Disadvantages:**
- Often in extreme environments
- Genome-centric focus (DNA repair, cancer suppression)
- Narrow insight into cellular resilience

450 — 60 million years of evolutionary distance

**Translational bottleneck**

Genome instability ≠ comprehensive aging biology

Misses proteostasis, metabolic/lipid homeostasis

Misses waste clearance & detoxification

Poor explanatory power for non-cancer aging diseases

Need for expanded model systems

**Figure 2. Model systems and comparative biology: defining the translational bottleneck in aging research.**

(A) Classical aging models and the translational bottleneck. Current short-lived experimental models offer rapid analysis, strong genetic tractability, and conserved pathways. However, their limited physiological complexity and large evolutionary distance from humans constrain predictive power, contributing to inconsistent clinical outcomes. These limitations underscore the need for expanded model systems that better capture human-relevant aging biology. (B) Comparative biology and the limitations of genome-centric paradigms. Exceptional long-lived species ("super agers") provide advantages such as complex tissues, conserved pathways, and naturally evolved resilience mechanisms. Yet many inhabit extreme ecological niches, and genome-centric interpretations of their longevity primarily emphasize DNA repair and cancer resistance. This narrow view overlooks critical cellular layers–including proteostasis, metabolic and lipid homeostasis, and cellular maintenance–that are central to human age-related dysfunctions such as neurodegeneration, metabolic decline, tissue toxicity, and loss of regenerative capacity. More integrative model systems are required to bridge this gap.

and a single FOXO ortholog (DAF-16) (Murphy and Hu, 2013; Lambie, 2002), constraining its ability to model mammalian stem-cell dynamics and endocrine regulation. Lifespan-extending mutations in *C. elegans* and also in Drosophila melanogaster (~700–800 million years of divergence) often show attenuated effects in mammals, reflecting the greater complexity of mammalian insulin/IGF-1 regulation (Sell, 2015). Similarly, interventions such as resveratrol reduce oxidative stress; however, ROS appear to play more critical roles in flies than mammals that possess more robust mechanisms to buffer ROS (Popa-Wagner et al, 2013; Santos et al, 2018). Importantly, although high levels of ROS can damage cellular structures, antioxidant supplementation has shown limited or even adverse effects in humans, as low and transient ROS-inducing signals function as adaptive stress responses that activate protective pathways (Ristow, 2014).

Short-lived vertebrates such as the African turquoise killifish (~450 million years of divergence), besides lacking several cell types of mammals (Kim et al, 2016), compress decades of mammalian lifespan and aging into months, altering the tempo of macromolecular and organelle turnover, as well as damage accumulation in certain tissues, influencing the fidelity of mimicking the aging pathological mechanisms that occur in long-lived species (Kim et al, 2016).

Consistent with this, mouse–human comparisons reveal modest overlap in age-associated transcriptomic changes across homologous tissues and substantial differences in telomerase biology and proteome stability (Matsuda et al, 2024; Palmer et al, 2021; Jones-Weinert et al, 2025).

These differences do not diminish the value of model organisms or the conserved longevity pathways discovered through them. Rather, they highlight that such pathways are executed within species-specific cellular and organelle environments

that shape physiological responses and the effectiveness of interventions in human tissues. Accordingly, strategies optimized in short-lived systems may not fully capture the demands placed on cells and tissues in long-lived species.

The translational challenge arises not from a lack of conserved biology, but from understanding how shared and species specific mechanisms function across different temporal scales, tissue architectures, and metabolic contexts. Importantly, classical model organisms remain indispensable for identifying core mechanisms of aging; the goal is not to replace them, but to complement them with mammalian and primate-based approaches that more closely reflect the cellular and physiological environments in which human aging occurs.

### Constraints of current comparative biology

Across mammals, lifespan and healthspan range from 2 to 70 years, or even longer (Tyshkovskiy et al, 2023). In the primate order, the lifespan differences span from 10 to 60 years (Huber et al, 2025). Comparative biology has begun to address the scope of these differences and evolutionary adaptations to long-lifespan on a molecular level through cross-species analyses (Rechsteiner et al, 2025; Tyshkovskiy et al, 2023), particularly in long-lived mammals such as bats, whales, elephants, and naked mole-rats (Sulak et al, 2016; Foley et al, 2018; Chen et al, 2025; Firsanov et al, 2025). These studies convincingly show that nature provides blueprints for resilience. Yet most comparative efforts on "super-ages" remain anchored in genome stability, DNA repair efficiency, and tumor resistance (Fig. 2B). This narrow mechanistic focus is also characteristic of studies on centenarians and the underlying intrinsic cellular mechanisms that contribute to their longevity (Garagnani et al, 2021).

Indeed, genomic and transcriptomic signatures alone, which poorly correlate

with proteome, lipidome, and metabolome, cannot fully explain how some mammals maintain mitochondrial efficiency, lipid and proteome homeostasis, lysosomal function, or redox balance far longer than others to resist aging. To illustrate further, in humans, the individuals with germline mutations in DNA polymerase ε (POLE) show a 2–9-fold increase in somatic mutations and a predisposition to cancer but lack overt progeroid phenotypes, underscoring that genomic instability alone cannot explain the biology of aging or of resilience (Robinson et al, 2021)

### Organelles as loci of healthy aging

Aging is fundamentally a problem of cellular homeostasis: the ability of cells to maintain energy balance, redox equilibrium, macromolecular turnover, waste cleanup, and membrane integrity over time (López-Otín et al, 2023) (Fig. 3). These processes are governed by the organelles that execute them——mitochondria, lysosomes, and peroxisomes, and their intrinsic molecular network. Many age-associated dysfunctions emerge at the level of cellular organelles, whose long-term stability underpins tissue function. Although being critical basic units of eukaryotic or more specifically, mammalian cells, initial comparative experiments reveal fundamental differences between cellular organelles of short to long- lived mammals as a result of evolutionary divergence to humans.

### Mitochondria: metabolic efficiency and damage control across lifespan scales

Mitochondria are central to healthspan and lifespan. Beyond energy, they generate key precursors for macromolecules, including lipids, nucleotides, and amino acids, as well as managing potentially harmful by-products such as reactive oxygen species (ROS), lactate, ammonia, and hydrogen

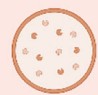

## Mitochondria:

### Core functions

- ATP production and metabolic flux coordination
- Redox signaling and ROS and toxic byproducts management
- Synthesis of key precursors (amino acids, nucleotides, lipids)

### Aging-related decline

- Reduced oxidative phosphorylation and ATP output
- Increased ROS leakage and redox imbalance
- Tissue-wide impact (cardiovascular, immune, nervous system)

### Species specific features

- Difference in speed of maturation, turnover, ROS generation and internal proteome (human versus mouse)

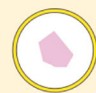

## Lysosomes:

### Core functions

- Degradation of damaged proteins, lipids, organelles
- Control of autophagy, nutrient signaling,
- Membrane repair and recycling of metabolites

### Aging-related decline

- Accumulation of undegraded substrates
- Dysfunction spreads to other organelles (mitochondria, ER)
- Tissue-wide impact (nervous immune, digestive system)

### Species specific features

- Difference in enzyme activities, speed of proteostasis and internal proteome (human versus mouse)

## Peroxisomes:

### Core functions

- Plasmalogen synthesis, lipid remodeling, membrane protection
- β-oxidation of very-long-chain fatty acids (VLCFA)
- Detoxification of ROS and $H_2O_2$

### Aging-related decline

- Reduced plasmalogen production
- Reduced detoxification of toxic lipids and oxidative species
- System-level effects (nervous system)

### Species specific features

- Difference in control organelle multiplication (cancer susceptibility), Multiple human disorders cannot be modeled in mice

**Figure 3.  Organelle-specific functions, aging-related decline, and species-dependent features shaping cellular resilience.**

Organelle function defines cellular resilience and health. However, aging perturbs these systems in organelle-specific ways, leading to cellular and tissue dysfunction. For long-lived lived animals, including mammals, it is essential that this decline does not appear prematurely and at the speed of the short-lived ones—indicating fundamental differences in maintenance and resilience of these critical cellular units, refined through millions of years of evolution. These mechanisms are underexplored in systematic comparative analysis, and represent an untapped avenue for therapeutic targeting in the prolongation of cellular and tissue health.

sulfide ($H_2S$) (Spinelli and Haigis, 2018). Mitochondria also serve as signaling platforms, regulating redox status, calcium homeostasis, and apoptotic pathways (Spinelli and Haigis, 2018). Despite encoding just 13 proteins in their own genome, mitochondria incorporate over 1000 nuclear-encoded proteins whose composition varies across tissues to match specific metabolic demands (Rath et al, 2021). Human and mouse mitochondria differ markedly in proteome composition (Pagliarini et al, 2008), import kinetics and poor cross-species mitochondrial transfer (Brestoff et al, 2025), ROS production rates, which are higher in mice (Adelman et al, 1988), maturation and turnover tempo (Iwata et al, 2023), reflecting evolutionary divergence. Such a mismatch explains why mitochondrial interventions developed in fast-aging species often fail to translate to human physiology.

## Lysosomes: sustaining proteostasis and waste clearance

Lysosomal efficiency is another key determinant of longevity. Lysosomes govern macromolecule recycling, membrane repair, nutrient signaling, stress resistance, and proteostasis (Mutvei et al, 2023). In aging, lysosomes undergo apparent structural and molecular changes, such as a change in size and number, a change in luminal pH, lysosomal membrane permeabilization, accumulation of undigested material, and upregulation of lysosomal enzymes such as β-galactosidase (Tan and Finkel, 2023). Age-related functional lysosomal decline not only affects the organelle itself but also other organelles and the whole cell, leading to a variety of age-related diseases (Tan and Finkel, 2023; Hughes and Gottschling, 2012). Human and mouse lysosomes differ in specific lipid degradation kinetics (Groener et al, 2000), protein degradation rate (Matsuda et al, 2024), and organellar proteome (Akter et al, 2023). More, humans develop age-related neurodegenerative diseases through lysosomal dysfunction that mice don't (Nixon et al, 2008), unless

genetically modified. Because long-lived mammals must sustain lysosomal functionality for many decades, they may have evolved structural and biochemical optimizations not present in rodents, and potentially in other short-lived species.

## Peroxisomes: lipid detoxification and membrane maintenance

Peroxisomes are indispensable to β-oxidation of very-long-chain fatty acids, preventing lipotoxicity (Lodhi and Semenkovich, 2014), supplying acetyl-CoA (Pietrocola et al, 2015), detoxification of hydrogen peroxide (Digiovanni et al, 2025; Fransen et al, 2017), and synthesis of plasmalogens that stabilize cellular membranes and act as endogenous antioxidants (Braverman and Moser, 2012). These functions are critical to neuroprotection and membrane integrity (Uzor et al, 2020). Aging affects these functions, causing a decline in ability to buffer ROS due to impaired catalase import (Legakis et al, 2002), and a decline in membrane integrity maintenance due to decreased plasmalogen synthesis, causing brain aging and neurodegeneration (Uzor et al, 2020). Although currently a comprehensive direct comparison between species is lacking, it is known that human peroxisomal multiplication is more tightly controlled compared to mice, preventing related cancerogenesis in the liver (Yang et al, 2008). The evolutionary differences are further indicated by many studies where mouse models failed to recapitulate human pathologies originating from peroxisomal dysfunction (Kocherlakota et al, 2023).

## From organelle divergence to comparative necessity

Short- and long-lived species operate under fundamentally different chronological timescales across multiple levels of biological organization, including at the cellular and subcellular levels. In short-lived organisms, rapid growth, high turnover, and tolerance for cumulative damage is compatible with

compressed lifespans. Long-lived mammals, in contrast, must maintain organelle function with high fidelity and functionality for decades, thus requiring mechanisms that stabilize macromolecular assemblies, preserve membrane integrity, maintain metabolic equilibrium, and sustain detoxification capacity over extended chronological time. As a result, interventions optimized in species with accelerated metabolic tempo and rapid organelle turnover may not perform equally efficiently in systems where maintenance must be sustained for decades.

This distinction is often overlooked in aging research when the developmental stage is being conflated with chronological time. For example, traditional mouse–human comparisons often equate old mouse with old human (Mercken et al, 2017), despite a nearly 40× difference in elapsed chronological time. The same categories of damage, such as ROS-associated molecular damage on proteins or the DNA, accumulate within years in mice but over decades in humans (Mercken et al, 2017). The most plausible explanation is that long-lived mammals possess intrinsic organelle-level mechanisms that slow the kinetics of damage production, stabilize macromolecular assemblies and their efficiency, or maintain detoxification capacity over vastly longer timescales. Indeed, when comparing mice and humans, given the ~90 million-year evolutionary distance (Pervouchine et al, 2015), there has been ample time for the emergence and refinement of such mechanisms under evolutionary pressures. Importantly, these features are expected to reside within organelle proteomes, lipid compositions, and metabolic configurations that shape long-term functional stability and are not readily inferred from DNA sequence or transcriptomic comparisons alone.

Advancing this underexplored area requires systematic examination of organelle biology across mammals spanning a wide range of lifespans, with inclusion of species that are more closely aligned with human physiology or chronological lifespan. Such analyses enable identification of

both shared and species-specific organelle architectures associated with extended lifespan. Crucially, these comparisons depend on harmonized cellular systems, aligned developmental states, and tightly standardized experimental contexts that allow intrinsic organelle adaptations to be distinguished from physiological and environmental influences. Excitingly, boosting these identified mechanisms that were already evolutionarily validated in mammals that are closer to humans can provide a fresh route for the development of next-generation anti-aging therapies with higher clinical success.

## The Comparative Metabolic Longevity Cell Atlas (CMLCA): an emerging framework to dissect organelle-level longevity

The Comparative Metabolic Longevity Cell Atlas (CMLCA) is based on the premise that longevity must be examined through direct, standardized measurement of organelle architecture and resilience in matched cell types derived from species spanning the mammalian lifespan spectrum, including humans. Such systematic comparisons enable identification of the evolutionary refined, organelle-level mechanistic nodes that sustain cellular stability over extended lifespans (Fig. 4A,B; core concept CMLCA).

Induced pluripotent stem cells (iPSCs) provide a scalable platform for this comparative approach. Biobanking iPSCs from

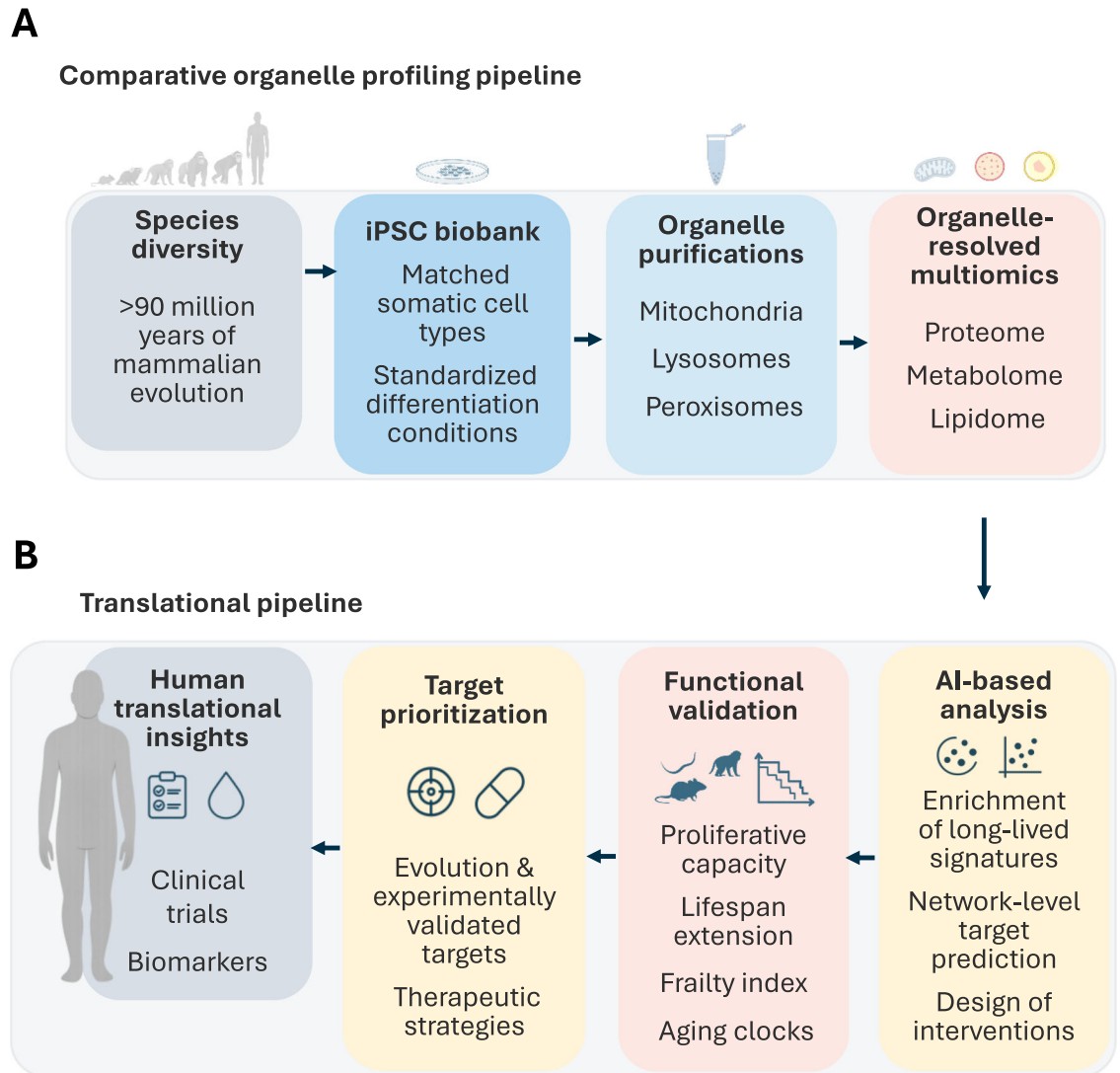

**Figure 4.  From comparative organelle profiling to translational target discovery in aging.**

(A) Within the Comparative Metabolic Longevity Cell Atlas (CMLCA), a diverse iPSC biobank spanning >90 million years of mammalian evolution enables standardized differentiation into matched somatic cell types. Purification of mitochondria, lysosomes, and peroxisomes, followed by organelle-resolved multi-omics, generates high-resolution profiles of conserved and long-lived–specific biological features. (B) Organelle-derived signatures that scale with lifespan guide target prioritization and inform therapeutic hypotheses. AI-based analysis integrates multi-omic and cross-species data to identify enriched longevity network nodes, enabling target identification and guiding the design of novel interventions. Functional validation across model systems evaluates effects on senescence, frailty indices, lifespan, and aging-clock dynamics. These outputs support the development of experimentally validated targets, therapeutic strategies, and translational pathways, including biomarker panels and clinical trial design.

diverse mammals, including short-lived rodents, intermediate-lived species, and long-lived primates, subsequently differentiating them into equivalent somatic cell types under harmonized conditions, allows intrinsic organelle properties to be examined independent of organism-level variability. The expandability of iPSCs further enables the generation of sufficient material for high-resolution downstream analyses.

To interrogate organelle architecture directly, the next step represents mitochondrial, lysosomal, and peroxisomal isolation from the iPSC-derived somatic cells coupled with multi-layered profiling——proteomics, lipidomics, and metabolomics——under basal and specific stress conditions. This approach provides a multidimensional view about organelle proteome, metabolome and both structural or free lipidome steady state composition, as well as dynamics and remodeling under the challenge, such as ROS induced stress or uncoupling, or proliferation induced senescence (Fig. 4A). Ensuring organelle purity (mock isolations and whole cell analysis), integrity, and reproducibility are essential for ensuring that downstream proteomic, lipidomic, and metabolomic measurements reflect true biological differences.

By standardizing cellular context and growth conditions, CMLCA creates a controlled comparative system in which intrinsic organelle properties can be assessed without confounding influences of physiology, environment, diet, or age. Mapping these features onto species lifespans enables discrimination between conserved resilience mechanisms and lineage-specific innovations. Some organelle traits may represent deep evolutionary solutions shared among long-lived mammals despite phylogenetic distance, whereas others may reflect species-specific adaptations, such as distinctive mitochondrial detoxification strategies or primate-specific molecular assemblies.

This framework enables a new set of questions: which organelle architectures repeatedly emerge in long-lived species? Which biochemical networks preserve organelle function across extended lifespans? Do long-lived mammals share characteristic stress-response signatures at the organelle level? Addressing these questions generates mechanistic hypotheses that can be experimentally tested by modulating candidate traits and assessing their effects on senescence and cellular resilience.

The resulting datasets of organelle proteomes, lipidomes, and metabolic profiles will enable computational modeling of resilience, identification of conserved molecular modules, AI-assisted prediction of longevity features, and cross-validation in primary cells, tissues, and in vivo models (Fig. 4B). Over time, the atlas will expand to include additional species, cell types, and perturbations——including stresses that model metabolic aging, inflammation, proteostasis decline, and environmental challenges——creating a continuously evolving reference for organelle-centered longevity biology.

Organelle-resolved multi-omic comparisons are expected to reveal recurring patterns associated with long lifespan. Lipidomics may uncover membrane compositions more resistant to oxidative injury; proteomics may identify unusually stable macromolecular assemblies and damage-handling systems to maintain functionality; metabolomics may highlight detoxification strategies and protective cofactor configurations enriched in long-lived mammals. Machine-learning approaches can prioritize not only single but also multi-component combinations of lipids, proteins, and metabolites—— "resilience modules"——for experimental validation in vitro and in vivo. Many of these mechanisms may not require genomic intervention but instead involve modulation of metabolic flux, restoration of lipid composition, or adjustment of redox balance, by targeted supplementation of molecular components optimized for safety and efficacy. Together, these readouts would define entry points for therapeutic design.

To further facilitate both discovery and translation, CMLCA is conceived as an open, collaborative atlas that complements genomic and transcriptomic resources by providing an organelle-resolved systems analysis. By integrating comparative biology with organelle-resolved systems analysis, the atlas aims not only to deepen the mechanistic foundation of aging research but also to enable a new class of gerotherapeutics with improved likelihood that discoveries translate to human healthspan (Fig. 4B).

While organelle-resolved multi-omics provides critical insight into molecular composition and metabolic state, resilience is also shaped by structural organization and spatial context. Cellular architecture influences enzyme behavior, organelle morphology, and bioenergetic properties in ways not fully captured by bulk biochemical analyses. For example, cytoskeletal organization can modulate the localization and kinetic behavior of metabolic enzymes, and mitochondrial length, network dynamics, and membrane potential vary according to cellular context and stress state. Such structural parameters influence damage susceptibility, redox balance, and metabolic flux, thereby contributing to functional resilience and health across lifespan scales. Accordingly, CMLCA is conceived not only as a molecular atlas but as a structural–functional platform that can integrate high-resolution imaging, quantitative morphology, and spatially resolved analyses in addition with organelle-resolved omics. Advances in microscopy, automated image analysis, and AI-assisted pattern recognition now make it feasible to incorporate architectural features into comparative longevity biology, enabling a more realistic representation of how organelle systems maintain stability over decades.

Altogether, CMLCA provides a conceptual and experimental framework for uncovering organelle-level strategies that support extended lifespan in mammals. By framing aging at the level of organelle design – how cellular components maintain functional fidelity over decades – the approach enriches the field beyond a predominantly genome-centric paradigm toward the prospect of organelle-level engineering.

## Current status, challenges, and next steps

The foundational components of the CMLCA are already in place. The current biobank spans short-, mid-, and long-lived mammals, including several primate species. Standardized differentiation protocols have been optimized across several species to yield matched somatic cell types, with more species in the pipeline. In parallel, pilot studies have demonstrated the feasibility of isolating organelles from select species, and an initial scalable workflow has been established for generating the biomass required for organelle-resolved multi-omic analysis. These advances ensure that the CMLCA is not a speculative framework but an emerging, experimentally grounded platform positioned for systematic expansion.

With this foundation, the next phase centers on developing robust pipelines for organelle-resolved proteomics, lipidomics, and metabolomics. Because mitochondria, lysosomes, and peroxisomes require

substantial biomass for high-quality purification, scaling biomass production and harmonizing organelle isolation across species now represent the principal rate-limiting steps. Initial small-scale isolations have been successful. A major transition point will occur once the first cross-species organelle-omic datasets become available, enabling comparative normalization, pathway alignment, and the discovery of molecular signatures that differ predictably with lifespan.

Importantly, the CMLCA does not aim to recapitulate whole-organism physiology. Its strength lies in isolating evolutionarily encoded subcellular mechanisms under tightly standardized conditions, generating a mechanistic filter for pathways with a higher likelihood of relevance to human aging biology, thus providing a complementary, deep-resolution perspective to in vivo studies in classical models.

As organelle purification, multi-omic profiling, and computational integration mature, the CMLCA is positioned to evolve from a conceptual prototype into a scalable discovery engine. The resulting organelle atlas will offer the field its first systematic, evolution-informed map of resilience across mammalian lifespan scales.

## Summary box

CMLCA now contains a growing panel of mammalian iPSC and fibroblast lines and is entering the phase of organelle-resolved multi-omic mapping. The next milestones——scalable organelle purification, cross-species metabolic alignment, and AI-driven discovery——will transform the platform into a mechanistic atlas of resilience pathways in long-lived mammals. This focused, in vitro comparative approach offers a practical route toward identifying longevity mechanisms with a higher likelihood for translation into human biology.

## Concluding remarks

Comparative biology has long highlighted the extraordinary diversity of mammalian lifespan, yet only recently have we begun to appreciate how deeply this diversity is encoded at the level of organelle structure and function. Organelles, such as mitochondria, lysosomes, and peroxisomes, may hold evolutionary solutions for maintaining cellular stability over vastly different chronological timescales. These solutions are not readily derived from genome sequence or inferred metabolic pathways; rather, they are encoded in the organelle-intrinsic proteomic, lipidomic, and biochemical features that shape the long-term fidelity of organelle systems.

Organelle-resolved comparisons across mammals all the way to humans can provide a new interpretive framework for fundamental and translational aging biology, guiding the design of interventions aimed at enhancing organelle stability and with a higher probability of translation to human biology, complementing conventional approaches. By focusing on the subcellular architectures that underlie sustained homeostasis, it will become possible to define the molecular features that distinguish prolonged resilience, building new aging Hallmarks (López-Otín et al, 2023; Schmauck-Medina et al, 2022), ones that do not describe what erodes with aging, but how to resist it – The Hallmarks of Resilience to Aging.

Ultimately, by integrating comparative organelle biology with emerging tools in multi-omics, computational modeling, and cell engineering, we may be able to reconstruct key aspects of long-lived cellular states in human tissues. Our hope is that this framework provides a conceptual foundation for future studies, informing both basic science and the development of therapeutics aimed at promoting healthy aging. As with other major transitions in geroscience, progress will depend on multidisciplinary collaboration, but the opportunity is clear: evolution has already performed the experiment. Our task now is to read, interpret, and apply its results.

## Peer review information

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

## Acknowledgements

I thank Nicolo Sbardellati for his instrumental contributions to the establishment of the Comparative Metabolic Longevity Cell Atlas (CMLCA) under the supervision of DC. We are grateful to the Vienna Zoo and the Research Institute of Wildlife Ecology, University of Veterinary Medicine Vienna, for providing primary samples from long-lived species. I thank the contributing laboratories and institutional biobanks that provided cellular resources supporting the expansion of the comparative panel. I thank David M Sabatini for early discussions on organelle isolation strategies and for thoughtful feedback on conceptual aspects of the CMLCA framework. I acknowledge the Division of Endocrinology and Metabolism, Medical University of Vienna, for institutional and logistical support. The development of the CMLCA framework received partial bridge funding support from the MetAGE Excellence Cluster (FWF COE14, Austrian Science Fund). Further information on the CMLCA initiative is available at www.cikeslab.com/CMLCA. The author used an AI-assisted tool for language editing and icon generation; all scientific content, interpretations, and conclusions are the author's own.

## Author contributions

Domagoj Cikes: Conceptualization; Writing—original draft; Writing—review and editing.

## Disclosure and competing interests statement

The author declares no competing interests.

