## [Peer Review File · EMBO Molecular Medicine]

Organelle resilience as a comparative blueprint for longevity

Domagoj Cikes

Corresponding author: Domagoj Cikes (domagoj.cikes@meduniwien.ac.at)

Review Timeline:

Submission Date:	17th Dec 25
Editorial Decision:	2nd Feb 26
Revision Received:	24th Feb 26
Editorial Decision:	19th Mar 26
Revision Received:	20th Mar 26
Accepted:	31st Mar 26

Editor: Lise Roth

Transaction Report:

2nd Feb 2026

Dear Dr. Cikes,

Thank you for the submission of your Perspective to EMBO Molecular Medicine, and please accept my apologies for the delay in getting back to you, which is due to the time it took to secure referees during the holiday season. We have now received feedback from the experts who agreed to evaluate your manuscript.

As you will see from the reports below, they overall found the article interesting and well written. Referee #1 nevertheless makes several suggestions to improve the interest and impact of your work.

We would therefore welcome a revised version of your manuscript that would address these points.

Please attach a covering letter giving details of the way in which you have handled each of the points raised by the referees.

For the figures, please note:

1. If there are certain aspects of your figure drafts that are based upon assumptions or where the scientific data remains ambiguous, please add a comment so that we can work with you on an accurate depiction. Please ensure the directionality and nature of interactions is presented accurately.
2. If the figures or part of the figures have been adapted from a published figure, please add this information to the figure legend (e.g., 'Adapted from...' or 'Based on...').
3. If you use an image data base for scientific iconography (e.g., BioRender), please let us know if you have a license that allows for publication in an academic journal.

Looking forward to receiving your revised manuscript at your earliest convenience.

With kind regards,

Lise Roth

***** Reviewer's comments *****

Referee #1 (Remarks for Author):

Dr. Domagoj Cikes presented a perspective/opinion paper entitled 'Organelle resilience as a comparative blueprint for longevity'. The topic is important and thus deserves a perspective. Addressing below questions will improve the quality of the paper.

Major concerns

1. The current version of the ABSTRACT long and redundant, and it should be further polished by making the text concise yet information-rich.
2. The structure of the manuscript is not easy to follow. The author should make the perspective more concise yet information-rich.
3. The author should tone down the tongue in the writings throughout the paper. E.g., the 1st sentence in the Introduction section "Aging research has identified numerous pathways that extend lifespan in short-lived organisms, yet translation to humans remains limited and often unexpectedly inconsistent" was too strong and vague for this reviewer as the four cited references do not fully support this strong statement. If the author wanted to continue this sentence, then more high profile research papers with clinical data (not just review paper) should be presented; additionally, they author needs to provide 2-3 examples regarding this statements. A deep concern is there are many hallmarks of ageing in humans as well as mammalian

ageing pathways are originally discovered in simple model systems, and many of these pathways (obviously, not all the pathways) are conserved from *C. elegans* to mice and humans (like FOXO3/DAF-16 and mTOR pathways).

4. In the INTRODUCTION section, different hallmarks of ageing should be introduced as many of the hallmarks were mentioned afterwards (PMID: 36040386; PMID: 36599349).

5. The NAD⁺ section should be extended, and indeed NAD⁺ augmentation has been shown to improve organelle function, like muscle and kidney and the brain. Additionally, for "NAD⁺ precursors (NR, NMN) safely raise NAD⁺ levels 12 87 though show modest, context-dependent benefits in randomized trials", NAD⁺ should be defined, and more representative research papers and review papers should be cited (PMID: 33888596; PMID: 40926126). The very significant benefits of NAD⁺ to premature ageing should be included also (PMID: 24813611; PMID: 27732836; PMID: 37899683).

Minor concerns

6. As there is only one author, please change "We introduce the" (ABSTRACT section, and elsewhere) to "I introduce the".

7. *C. elegans* should be in italic.

8. Regarding ROS "ROS appear to play more critical roles in flies than mammals that possess more robust mechanisms to buffer ROS", it is important to mention that ROS can be good and bad - dose matters (PMID: 24999941).

9. For "The authors declare that they have no conflicts of interest.", please revise to singular.

Referee #2 (Remarks for Author):

Many aging research results have been obtained from experiments investigating effects on model organisms. A few of the effects can be transferred to human aging and are thus potentially significant for translational medicine. Recent studies focus on DNA sequence, repair pathways, transcriptional control and proteomic, lipidomic and metabolic features on cellular or on organ level. This essay points to organelle - level intrinsic resilience as a means to achieve longevity by sustaining cellular homeostasis over decades, i.e. by maintaining macromolecular turnover, energy and redox balance, and detoxification in humans. Evolution shaped the aging processes of the broad range of model organisms reaching from yeast, to *Caenorhabditis* or *Drosophila* to vertebrates with killifish, zebrafish, mice and mammals which are closer to humans as are non-human primates (marmottes, rhesus monkeys, squirrels, bats and others). This list shows that a very broad range of ecological conditions are covered by these organisms, each of them can tell us principles which support lifetime by sustained functionality of metabolic properties. But these properties are adaptations to very special external conditions not typical for humans. On the other hand, life processes share many basic evolutionary origins and therefore knowledge gained from one organism may well be translated to another one if we understand the mechanisms.

Many findings on experimental extension of longevity in small animals have failed to produce similar effects in humans. One of the most reliable factors increasing longevity, almost independent from the test system, is reduction of nutrition (fasting). This is also evident in the various forms of "suspended life" and makes hibernation in mammals so interesting for aging research. "Aging is fundamentally a problem of cellular homeostasis: the ability of cells to maintain energy balance, redox equilibrium, macromolecular turnover, waste cleanup and membrane integrity over time. These processes are governed by the organelles that execute them - mitochondria, lysosomes, and peroxisomes, and their intrinsic molecular network." This sentence in the manuscript of Domagoj Cikes very clearly and convincingly describes the research situation in which, beyond transcriptome, proteome, and lipidome analysis, the functionally essential cell structures are included in the considerations, thus making research more holistic. Only by taking into account the complexity of the object (in this case, the aging organism) will our understanding advance to the point where fundamental and translational aging medicine becomes possible. The solution is to create an organelle-based database in which the molecular foundations are processed in an organelle-specific manner: a "Comparative Metabolic Longevity Cell Atlas (CMLCA)". This approach is very well founded and convincingly developed in the present work. The starting material for experimental data collection is somatic cell culture (preferably 3D cultures) developed from stem cells using standardized differentiation procedures, which are derived from the respective test materials. This approach enables both cell-specific and organism-specific stem cell development and assessment of the aging status of the starting material. The evolutionary differences in genetic organization can thus be integrated into the assessment of aging. This also allows, for example, the mitochondrial-lysosomal axis to be systematically investigated. However, the database is still in development and has not yet been put to the test. Unfortunately, this research approach also fails to address the question of the significance of structures. This concerns, for example, the interaction of molecules beyond individual pathways, such as the interaction of cytoskeletal elements, especially actin, with glycolysis enzymes. The cytoskeletal organization of senescent cells differs in many ways from that of reproductive cells. Also cellular structure adds to information transfer exceeding DNA-related propagation of information. This cytoskeletal elements not only change the organization of glycolysis enzymes but also their biochemical properties (actin-associated enzymes no longer exhibit Michaelis-Menten behaviour). The interaction is mutual: the polymerization behaviour of the cytoskeletal elements is altered, as is the enzymatic activity of the enzymes. Another example of structures relevant to aging is the length of mitochondria, which has a major influence on the resilience of mitochondria against damaging influences (e.g., from ROS). In addition, mitochondrial electrochemical potential, a key parameter related to aging, may be different depending on the position of a mitochondrion within a cell. These are examples of the importance of structure that are not captured in standard biochemical analyses, microscopic structural investigations are necessary here.

At first glance, taking such a wealth of factors into account may seem overwhelmingly complex, but it brings us much closer to a realistic picture of the aging process and is also becoming practically feasible thanks to advances in data analysis (e.g., AI and

automation of biochemical analyses).

Response to Reviewer #1

I thank the reviewer for the thorough and constructive critique, which helped improve the manuscript's structure, tone, and scientific balance.

Major concerns

1. The abstract is long and redundant.

The abstract has been rewritten and shortened. Redundant phrasing was removed and the text was made more concise and information-rich. The revised version now clearly outlines the translational challenge, introduces organelle resilience as a conceptual framework, and positions the CMLCA platform as enabling infrastructure.

2. The manuscript structure is not easy to follow.

The manuscript was reorganized to improve clarity and flow. Paragraphs were tightened, overlapping content was removed, and transitions between sections were clarified. The revised structure now proceeds from:

- hallmarks of aging and translational challenges,
- current intervention landscape,
- limitations of existing model systems,
- organelle-level execution of homeostasis,
- introduction of the comparative organelle framework and CMLCA.

3. Tone should be moderated; statements on translation were too strong.

The tone was revised throughout the manuscript to be more analytical and balanced. Statements regarding translational limitations were reframed to emphasize variability across physiological contexts, tissues, and intervention strategies, and are now supported with representative clinical data rather than generalized statements. The Introduction was rewritten to acknowledge conservation of hallmarks of aging and longevity pathways across species and to incorporate clinical context where appropriate.

4. Hallmarks of aging should be introduced earlier.

The Introduction now explicitly introduces the hallmarks framework and situates the Perspective as extending this foundation by focusing on species-specific and organelle-level execution of homeostatic processes across long biological timescales.

5. NAD⁺ section should be expanded, defined, and include premature-aging evidence.

The NAD⁺ section was substantially revised and expanded.

- NAD⁺ is now defined as a central redox cofactor and signaling metabolite linking mitochondrial metabolism, DNA repair, autophagy/mitophagy, and inflammatory regulation.
- Evidence of benefits across tissues in preclinical systems was incorporated.
- Human data were expanded and presented in a balanced manner, emphasizing safety, biological activity, and context-dependent functional outcomes.
- Examples of benefits in premature-aging and disease contexts were included.
- Additional representative literature and reviews were integrated as suggested.

Minor concerns

6. Use singular author voice.

All instances were revised to singular phrasing.

7. *C. elegans* italicization.

Corrected.

8. ROS discussion should acknowledge dose dependence.

The ROS section was revised to reflect the dual role of ROS as both damaging agents at high levels and adaptive signaling mediators at low levels.

9. Conflict-of-interest statement should be singular.

Corrected.

Response to Reviewer #2

I thank the reviewer for the supportive evaluation and for highlighting important conceptual aspects that helped refine the manuscript.

Role of model organisms and evolutionary context

The manuscript was revised to more explicitly acknowledge the foundational role of model organisms in identifying conserved longevity pathways. The Perspective now emphasizes that the proposed comparative organelle framework complements, rather than replaces existing model systems by addressing differences in lifespan scale, tissue physiology, and metabolic context.

Nutrient restriction and fasting

A dedicated statement was added noting caloric restriction and fasting as among the most consistent modulators of lifespan across species and central to metabolic regulation in aging biology.

Organelle resilience as a conceptual framework

The manuscript further clarifies organelle resilience as an execution layer of aging biology that complements genomic and transcriptomic regulation and strengthens the positioning of CMLCA as enabling infrastructure for comparative analysis.

Structural organization and cellular architecture

In response to the reviewer's comments, the CMLCA section was expanded to explicitly address structural determinants of aging beyond molecular multi-omics. The revised text now highlights:

- the importance of spatial organization and cellular architecture,
- interactions between cytoskeletal structure and metabolic processes,
- the relevance of mitochondrial morphology and positioning, and

- the need to integrate imaging and structural analyses with molecular and functional readouts.

The manuscript now explicitly notes that advances in imaging, automation, and data integration make such approaches increasingly feasible.

Additional global revisions

Across the manuscript:

- language tightened and redundancy reduced,
- tone moderated,
- transitions strengthened,
- acknowledgments clarified,
- formatting corrections implemented.

I believe these revisions have significantly strengthened the manuscript's conceptual clarity and balance while preserving its central thesis and scientific intent. I am grateful for the reviewers' constructive feedback, which helped refine the framing and presentation of the work.

I hope that the revised manuscript is now suitable for publication.

Sincerely,

Domagoj Cikes

19th Mar 2026

Dear Dr. Cikes,

Thank you for the submission of your revised article to EMBO Molecular Medicine. Referee #1 has reviewed your revised manuscript and is satisfied with the revisions. I will therefore be able to accept your manuscript once the following editorial issues are addressed:

- Rename "Declaration of interests" to "Disclosure and Competing Interests Statement"
- Remove "Author Information" from the manuscript text; the information for Authors and Affiliations is already on the manuscript title page and Contributions should be listed in the submission system only.
- References list format should be corrected to alphabetical order and 10 author names listed before et al.
- You currently list 138 references. While we do not have strict limit for the number of references, we do recommend a maximum of 50 references for Perspectives, please adjust accordingly.
- Remove figure legends from the figure files and add them to the end of the manuscript text, after the References.

Looking forward to receiving your revised manuscript,

With kind regards,

Lise Roth

***** Reviewer's comments *****

Referee #1 (Remarks for Author):

Domagoj Cikes did a good job in addressing the questions raised by this reviewer and other reviewer(s) and the quality of the paper is much improved (including balanced citations and tuning down the tone of writing of the conclusions).

The authors addressed the remaining editorial issues.

31st Mar 2026

Dear Dr. Cikes,

We are pleased to inform you that your manuscript is accepted for publication and is now being sent to our publisher to be included in the next available issue of EMBO Molecular Medicine.

Your manuscript will be processed for publication by EMBO Press. It will be copy edited and you will receive page proofs prior to publication. Please note that you will be contacted by Springer Nature Author Services to complete licensing information. This Perspective article is free of charge.

With kind regards,

Lise Roth
